# Permutation-Based Rank Test in the Presence of Discretization and Application in Causal Discovery with Mixed Data

**Xinshuai Dong** [1] **Ignavier Ng** [1] **Boyang Sun** [2] **Haoyue Dai** [1] **Guang-Yuan Hao** [2]
**Shunxing Fan** [2] **Peter Spirtes** [1] **Yumou Qiu** [3] **Kun Zhang** [1 2]

## Abstract

Recent advances have shown that statistical tests for the rank of cross-covariance matrices play an important role in causal discovery. These rank tests include partial correlation tests as special cases and provide further graphical information about latent variables. Existing rank tests typically assume that all the continuous variables can be perfectly measured, and yet, in practice many variables can only be measured after discretization. For example, in psychometric studies, the continuous level of certain personality dimensions of a person can only be measured after being discretized into order-preserving options such as disagree, neutral, and agree. Motivated by this, we propose **M**ixed data **P**ermutation-based **R**ank **T**est (MPRT), which properly controls the statistical errors even when some or all variables are discretized. Theoretically, we establish the exchangeability and estimate the asymptotic null distribution by permutations; as a consequence, MPRT can effectively control the Type I error in the presence of discretization while previous methods cannot. Empirically, our method is validated by extensive experiments on synthetic data and real-world data to demonstrate its effectiveness as well as applicability in causal discovery (code will be available at `https://github.com/dongxinshuai/scm-identify`).

## 1. Introduction and Related Work

Recent advances have shown that the rank of a cross-covariance matrix and its statistical test play essential roles in multiple fields of statistics especially in causal discovery

---
[1]Carnegie Mellon University [2]Mohamed bin Zayed University of Artificial Intelligence [3]Peking University. Correspondence to: Kun Zhang <kunz1@cmu.edu>.

*Proceedings of the 42nd International Conference on Machine Learning*, Vancouver, Canada. PMLR 267, 2025. Copyright 2025 by the author(s).

(Sullivant et al., 2010; Spirtes, 2013). From one perspective, Independence and Conditional Independence (CI) are crucial concepts in causal discovery and Bayesian network learning (Pearl et al., 2000; Spirtes et al., 2000; Koller & Friedman, 2009) due to its relation to d-separations (Pearl, 1988), and it has been shown that rank tests take those linear CI tests as special cases (Sullivant et al., 2010; Di, 2009; Dong et al., 2024a). From another point of view, rank of a cross-covariance matrix corresponds to t-separations in a graph (Sullivant et al., 2010), which contain graphical information that can be used to identify latent variables (Huang et al., 2022; Dong et al., 2024a). A more detailed discussion about related work can be found in Appendix D.

Existing statistical rank tests (Anderson, 1984) are often built upon Canonical Correlation Analysis (CCA) (Jordan, 1875; Hotelling, 1992), with a likelihood ratio based test statistics. Despite their effectiveness, existing methods rely on the strong assumption that all the variables concerned can be perfectly measured. However, in many fields, it is often the case that the best available data are just discretized approximations of some underlying continuous variable (formally defined in Eq. 1). For example, in mental health, anxiety levels are often categorized into levels such as mild, moderate, or severe, according to some latent thresholds (Johnson et al., 2019). Examples can be found in multiple fields such as finance (Changsheng & Yongfeng, 2012), psychology (Lord & Novick, 2008), biometrics (Finney, 1952) and econometrics (Nerlove & Press, 1973), where continuous variables are often assumed to be observed as discretized values.

When discretization is present, existing rank tests can hardly work. The main reason lies is that the discretized values only reflect the order of the data, leading to cross-covariance estimates that may differ significantly from the underlying cross-covariance matrix (also illustrated in Figure 1). Furthermore, even though the true underlying cross-covariance matrix can be estimated by maximum likelihood-based methods such as polychoric and polyserial correlations (Olsson et al., 1982; Olsson, 1979), they cannot be directly plugged into existing rank tests. This is because the involved discretization and maximum likelihood processes change the distribution of

test statistics to a considerable extent and thus the p-values cannot be correctly calculated. As a consequence, Type I errors of existing methods cannot be effectively controlled. Both of these points are elaborated in Section 2.2.

To properly address the issue of discretization, in this paper, we propose a novel statistic rank test based on permutation, i.e., Mixed data Permutation-based Rank Test (MPRT) that can accommodate continuous, partially discretized, or fully discretized observations. Specifically, in the presence of discretization, the underlying cross-covariance can be estimated by maximum likelihood estimator, but the information loss resulting from discretization and the additional estimation steps make the derivation of the null distribution highly non-trivial. To this end, we start with the continuous case and establish exchangeability of linear projections of concerned variables (Theorem 4), based on which the null distribution can be empirically estimated by permutations. When some observations are discretized, the exchangeability still holds but we do not have direct access to permutable data. Fortunately, we show that the concerned statistic distribution can still be consistently estimated by properly using permuted discretized observations (Theorem 5). We summarize our key contributions as follows.

- To our best knowledge, we propose the first statistic rank test i.e., Mixed data Permutation-based Rank Test (MPRT), that properly deals with the problem of discretization. Rank test takes partial correlation CI test as a special case and thus the problem is crucial to many scientific fields such as psychology, biometrics, and econometrics, where discretizations are ubiquitous.

- Theoretically, we estimate the asymptotic null distribution by effectively making use of data permutations, and thus properly controls the Type I error. The setting considered is rather general: for the test of rank($\Sigma_{\mathbf{X},\mathbf{Y}}$), both $\mathbf{X}$ and $\mathbf{Y}$ are allowed to be either fully continuous, partially discretized, or fully discretized. Thus, our method also includes the fully-continuous rank test as a special case.

- Empirically, we validate our novel rank test under multiple synthetic settings where our method is shown to control Type I error properly and Type II error effectively, while existing methods cannot. We also use a real-world dataset to show the practicability of the proposed rank test and illustrate its application in causal discovery.

## 2. Preliminaries

### 2.1. Problem Setting

Suppose that we have a set of $M$ observed random variables $\mathbf{V} = \{\mathsf{V}_j\}_{j=1}^M$ that are jointly Gaussian. However, for some of these variables, direct observations are unavailable. We use $\mathbb{C}_{\mathbf{V}}$ and $\mathbb{D}_{\mathbf{V}}$ to denote the index set of those variables in $\mathbf{V}$ that we have direct observations and that of those we

only have order-preserving discretized observations, respectively. Assume that we have $N$ i.i.d., observations of these variables. The underlying true data matrix is $\boldsymbol{D} \in \mathbb{R}^{N \times M}$, while we only have access to $\tilde{\boldsymbol{D}}$, where some columns are discretized. Specifically, for $j \in \mathbb{C}_{\mathbf{V}}$, $\tilde{\boldsymbol{D}}_{:,j} = \boldsymbol{D}_{:,j}$, while for those $j \in \mathbb{D}_{\mathbf{V}}$, the observations are discretized in the following fashion:

$$\tilde{\boldsymbol{D}}_{i,j} = t, \text{ if } T_t^j < \boldsymbol{D}_{i,j} \le T_{t+1}^j, \\ \text{for } i \in \{1, ..., N\}, t \in \{1, ..., C_j\}, \quad (1)$$

where $C_j$ is the cardinality of the domain of the discretized observation of $\mathsf{V}_j$, $T_t^j$ refers to the $t$-th threshold for variable $\mathsf{V}_j$, $T_1^j \triangleq -\infty$, and $T_{C_j+1}^j \triangleq \infty$.

We are interested in the rank of the population cross-covariance matrix over certain combinations of variables, e.g., $\Sigma_{\mathbf{X},\mathbf{Y}}$, where $\mathbf{X} \subseteq \mathbf{V}$ and $\mathbf{Y} \subseteq \mathbf{V}$ ($\mathbf{X}$ and $\mathbf{Y}$ are not necessarily disjoint). The rank information is crucial to causal discovery (Spirtes et al., 2000) and will be detailed in Section 2.2. Ideally, we would expect that we have infinite datapoints and there is no discretization; in this case, the sample covariance $\hat{\Sigma}_{\mathbf{X},\mathbf{Y}}$ would be exactly the same as the population covariance, and the rank can be easily calculated by linear algebra. However, in practice we only have finite datapoints and for some of the variables we only have discretized observations. Thus, it is crucial to consider the following problem: in the finite sample case and in the presence of discretization, we only have access to $\tilde{\boldsymbol{D}}$ instead of $\boldsymbol{D}$, how to build a valid statistic test that properly controls the Type I error for testing the rank of a cross-covariance matrix $\Sigma_{\mathbf{X},\mathbf{Y}}$?

### 2.2. Why this Problem is Important?

In this section we will briefly discuss why rank test is important in the context of causal discovery as well as why it is crucial to deal with discretization.

#### Rank Test Takses Linear CI Test as a Special Case

In causal discovery, we aim to find the underlying causal graph among variables given observational data. The most classical approach is to use conditional independence (CI) relationships to identify d-separations in a graph; see, e.g., the PC algorithm (Spirtes et al., 2000). This idea is captured by the following theorem.

**Theorem 1** (Conditional Independence and D-separation (Pearl, 1988))**.** *Under the Markov and faithfulness assumption, for disjoint sets of variables* $\mathbf{A}$*,* $\mathbf{B}$ *and* $\mathbf{C}$*,* $\mathbf{C}$ *d-separates* $\mathbf{A}$ *and* $\mathbf{B}$ *in graph* $\mathcal{G}$*, iff* $\mathbf{A} \perp\!\!\!\perp \mathbf{B}|\mathbf{C}$ *holds for every distribution in the graphical model associated to* $\mathcal{G}$*.*

In practice, we often consider linear causal models where the CI test can be done by e.g., Fisher-Z (Fisher et al., 1921). It has been shown that, for linear causal models,

d-separations between variables can also be uncovered by rank tests, which is summarized in the following theorem.

**Theorem 2** (D-separation by Rank Test (Dong et al., 2024a)). *Suppose a linear causal model with graph $\mathcal{G}$ and assume rank faithfulness (Spirtes, 2013). For disjoint variable sets $\mathbf{A}$, $\mathbf{B}$, and $\mathbf{C}$, we have $\mathbf{C}$ d-separates $\mathbf{A}$ and $\mathbf{B}$ in graph $\mathcal{G}$, if and only if $rank(\Sigma_{\mathbf{A}\cup\mathbf{C},\mathbf{B}\cup\mathbf{C}}) = |\mathbf{C}|$.*

The above Theorem 2 says that d-separations can also be inferred from rank of a cross-covariance matrix, and thus for causal discovery of linear causal models, partial correlation test / linear CI test can be substituted by rank test.

**Rank Relates to T-sep that Indicates Latent Variables**

Next, we show that rank of cross-covariance informs something beyond d-separations. Specifically, t-separations (Sullivant et al., 2010) can be inferred from rank, and t-separations can be used to identify latent variables. The relation between rank and t-separations is given as follows.

**Theorem 3** (Rank and T-separation (Sullivant et al., 2010)). *Given two sets of variables $\mathbf{A}$ and $\mathbf{B}$ from a linear model with graph $\mathcal{G}$ and assume rank faithfulness. We have:*

$$rank(\Sigma_{\mathbf{A},\mathbf{B}}) = \min\{|\mathbf{C_A}| + |\mathbf{C_B}| : (\mathbf{C_A}, \mathbf{C_B}) \atop t\text{-separates } \mathbf{A} \text{ from } \mathbf{B} \text{ in } \mathcal{G}\}, \quad (2)$$

*where $\Sigma_{\mathbf{A},\mathbf{B}}$ is the cross-covariance over $\mathbf{A}$ and $\mathbf{B}$.*

The left-hand side of Equation 2 is about properties of the observational distribution, while the right-hand side describes properties of the graph. An example highlighting the greater informativeness of rank compared to CI is as follows. Consider the graph $\mathcal{G}$ in Figure 5, where $\{X_1, X_2\}$ and $\{X_3, X_4\}$ are d-separated by $L_1$, but we can never infer that from any CI test, i.e., we can never check whether $\{X_1, X_2\} \perp\!\!\!\perp \{X_3, X_4\}|L_1$ holds, as $L_1$ is not observed. In contrast, using rank information, we can infer that $rank(\Sigma_{\{X_1 X_2\},\{X_3 X_4\}}) = 1$, which implies $\{X_1, X_2\}$ and $\{X_3, X_4\}$ are t-separated by one latent variable. The rationale behind is that the t-separation of two set of variables $\mathbf{A}$, $\mathbf{B}$ by $(\mathbf{C_A}, \mathbf{C_B})$ can be inferred through rank, without actually observing any element in $(\mathbf{C_A}, \mathbf{C_B})$. A more detailed discussion can be found in (Dong et al., 2024a).

**Discretization is Ubiquitous and Needs to be Handled**

Discretization is ubiquitous in many scientific fields. For instance, it is common to come across concepts that cannot be measured directly, such as depression, anxiety, attitude, and the observations of such variables are often the result of coarse-grained measurement of the underlying continuous ones. More examples can be found in fields like psychology (Lord & Novick, 2008), biometrics (Finney, 1952) and econometrics (Nerlove & Press, 1973), where it is widely accepted to assume a continuous variable underlies a dichotomous or polychotomous observed one.

In the context of rank test, what should we do to deal with such a ubiquitous discretization problem? One naive way is to just treat these ordinal values as continuous ones and test the rank of a cross-covariance matrix as usual, and yet it cannot work. The reason lies in that the observed values of these discretized variables just represent the ordering and the values can be rather arbitrary. For example, assume that the original continuous observations are discretized into three levels represented by $\{1, 2, 3\}$ respectively; one can alternatively uses $\{1, 2, 2.1\}$ or $\{1, 2, 10^{16}\}$ to represent the three levels. If we directly use the ordinal values, the resulting cross-covariance matrix can be very different from the ground truth one, leading to meaningless results. An example can be found in Figure 1, where (a) shows the population cross-covariance and (b) shows the counterpart calculated by using discretized observations. Even with infinite samples, the two matrices are totally different, and the rank of the matrix in (a) is 1 while rank of that in (b) is 3. Next, we will show that, even if we can use maximum likelihood to estimate the correlation first, the problem is still highly non-trivial.

**2.3. Classical Rank Test with Estimated Correlation**

We have shown that the naive solution of directly using the ordinal values cannot work. Thus, one may wonder another straightforward one - estimate the correlations first (which can be done by maximizing likelihood, detailed in Section 3.3), and then plug the estimated correlations into a standard CCA rank test. In this section we will show that this straightforward solution cannot work either; more specifically, the Type-I errors cannot be effectively controlled.

We start with a brief introduction to the classical rank test, which is based on Canonical Correlation Analysis (CCA) (Jordan, 1875; Hotelling, 1992). The key design of a test typically is to find a suitable statistic and to derive its distribution under the null hypothesis. As for rank test of cross-covariance $\Sigma_{\mathbf{X},\mathbf{Y}}$, statistics based on CCA scores between $\mathbf{X}$ and $\mathbf{Y}$ are found to be very effective. For $|\mathbf{X}| = P, |\mathbf{Y}| = Q$, and $K = \min(P, Q)$, the CCA problem is as follows:

$$\max_{\mathbf{A}\in\mathbb{R}^{P\times K}, \mathbf{B}\in\mathbb{R}^{Q\times K}} \text{tr}(\mathbf{A}^T\hat{\Sigma}_{\mathbf{X},\mathbf{Y}}\mathbf{B}),$$
$$\text{s.t., } \mathbf{A}^T\hat{\Sigma}_{\mathbf{X}}\mathbf{A} = \mathbf{B}^T\hat{\Sigma}_{\mathbf{Y}}\mathbf{B} = \mathbf{I}. \quad (3)$$

Assume that the solution to Eq. 3 leads to CCA scores between $\mathbf{X}$ and $\mathbf{Y}$ as $\{r_i\}_{i=1}^K$. With the null hypothesis that $rank(\Sigma_{\mathbf{X},\mathbf{Y}}) \leq k$, referred to as $\mathcal{H}_0^k$, we would expect that the top-$k$ CCA scores are non-zero and the rest ones are all zero. This leads to a likelihood-ratio-based test statistics (Anderson, 1984) under $\mathcal{H}_0^k$ as follows.

$$\lambda_k = -\left(N - \frac{P+Q+3}{2}\right)\ln(\Pi_{i=k+1}^K(1-r_i^2)), \quad (4)$$

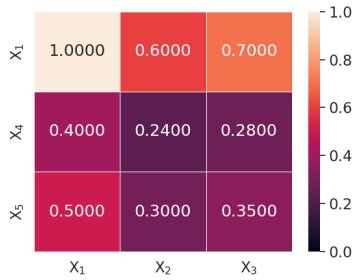 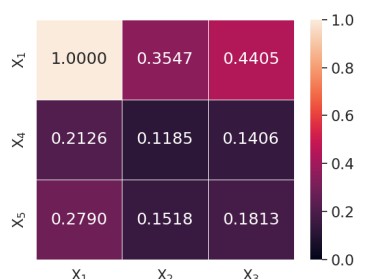 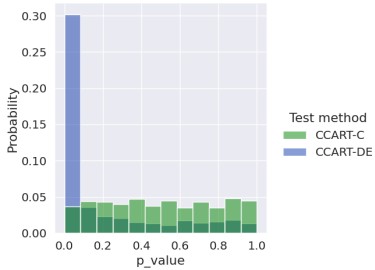

(a) Population cross-covariance matrix over continuous variables.

(b) Cross-covariance matrix using discretized data with $N \to \infty$.

(c) Distributions of p-values of CCART-C and CCART-DE.

*Figure 1.* Subfigures (a) and (b) together show we cannot directly take the discrete values for the calculation of rank of the covariance. Subfigure (c) shows that directly plugging an estimated cross-covariance into a rank test does not work as Type I cannot be controlled.

which has been shown to approximately follow a chi-square distribution with degree of freedom $(P - k + 1)(Q - k + 1)$. To perform the rank test, one only has to calculate $\lambda_k$ and the related chi-square distribution to get the p-value.

In Eq 3, $\hat{\Sigma}_{\mathbf{X},\mathbf{Y}}$ refers to the sample covariance $\frac{\mathbf{D}^{\mathbf{X}^T}\mathbf{D}^{\mathbf{Y}}}{N-1}$. In the presence of discretization, we only have access to $\tilde{\mathbf{D}}^{\mathbf{X}}$ and $\tilde{\mathbf{D}}^{\mathbf{Y}}$, but we can still estimate the cross-correlation by maximizing the likelihood (detailed in Section 3.3), and take the estimation into Eq. 3 to calculate the CCA scores and thus the test statistics. However, due to the information loss introduced by discretization and the additional maximum likelihood steps, the distribution of the statistics is changed to a considerable extent. An example is shown in Figure 1 (c), where CCART-C refers to CCA rank test using the original continuous observations and CCART-DE refers to first estimating the correlations by maximum likelihood using discrete data and then plugging it into the CCA rank test. As shown, the p-values of CCART-C are uniformly distributed while the p-values of CCART-DE are clearly not; most of them are near to zero and thus the test tends to reject everything, leading to unacceptably large Type I errors (also validated in Section 4.2 and Figure 2).

Ideally, we would expect to derive the updated distribution of the statistics, and yet the involved likelihood maximization steps make it very difficult. Therefore, we aim to solve this problem by estimating the empirical cdf of the null distribution using permutations, detailed in what follows.

## 3. Mixed Data Permutation-based Rank Test

In this section, we propose MPRT. A brief introduction to permutation test can be found in Appendix C.1. We start with the all continuous case.

### 3.1. All Continuous Case

Assume that we are interested in the rank of $\Sigma_{\mathbf{X},\mathbf{Y}}$, where $|\mathbf{X}| = P$ and $|\mathbf{Y}| = Q$ and their corresponding data matri-

ces are $\tilde{\mathbf{D}}^{\mathbf{X}} \in \mathbb{R}^{N \times P}$ and $\tilde{\mathbf{D}}^{\mathbf{Y}} \in \mathbb{R}^{N \times Q}$ respectively. The first crucial step is to solve the CCA problem defined in Eq 3, by Singular Value Decomposition (SVD) as follows.

$$\mathbf{U}\mathbf{S}\mathbf{V} = \hat{\Sigma}_{\mathbf{X}}^{-\frac{1}{2}}\hat{\Sigma}_{\mathbf{X},\mathbf{Y}}\hat{\Sigma}_{\mathbf{Y}}^{-\frac{1}{2}},$$
$$\mathbf{A} = \hat{\Sigma}_{\mathbf{X}}^{-\frac{1}{2}T}\mathbf{U} \text{ and } \mathbf{B} = \hat{\Sigma}_{\mathbf{Y}}^{-\frac{1}{2}T}\mathbf{V}^T, \quad (5)$$

where $\mathbf{A}$ and $\mathbf{B}$ are two linear projection matrices and the two CCA variables are $\mathbf{C}_{\mathbf{X}} = \mathbf{A}^T\mathbf{X}$ and $\mathbf{C}_{\mathbf{Y}} = \mathbf{B}^T\mathbf{Y}$. $\mathbf{C}_{\mathbf{X}}$ and $\mathbf{C}_{\mathbf{Y}}$ have two good properties: (i) $\hat{\Sigma}_{\mathbf{C}_{\mathbf{X}}} = \hat{\Sigma}_{\mathbf{C}_{\mathbf{Y}}} = \mathbf{I}$, and $\hat{\Sigma}_{\mathbf{C}_{\mathbf{X}},\mathbf{C}_{\mathbf{Y}}}$ is a diagonal matrix; (ii) under null hypothesis $\mathcal{H}_0^k : \text{rank}(\Sigma_{\mathbf{X},\mathbf{Y}}) \leq k$, only the top-k diagonal entries of $\Sigma_{\mathbf{C}_{\mathbf{X}},\mathbf{C}_{\mathbf{Y}}}$ are nonzero and the rest of the diagonal entries should be zero. Taking these two into consideration, we have the exchangeability between $\mathbf{C}_{\mathbf{X}k:}$ and $\mathbf{C}_{\mathbf{Y}k:}$, which is formalized in the following Theorem 4 (proof of which can be found in Appendix).

**Theorem 4** (Exchangeability of $\mathbf{C}_{\mathbf{X}k:}$ and $\mathbf{C}_{\mathbf{Y}k:}$)**.** *Given a set of variables* $\mathbf{V}$ *that are jointly gaussian, under null hypothesis* $\mathcal{H}_0^k : \text{rank}(\Sigma_{\mathbf{X},\mathbf{Y}}) \leq k$*, where* $\mathbf{X}, \mathbf{Y} \subseteq \mathbf{V}$*, random vectors* $\mathbf{C}_{\mathbf{X}k:}$ *and* $\mathbf{C}_{\mathbf{Y}k:}$ *are asymptotically independent with each other.*

Based on the exchangeability between $\mathbf{C}_{\mathbf{X}k:}$ and $\mathbf{C}_{\mathbf{Y}k:}$, we can permute the data matrix of $\mathbf{C}_{\mathbf{X}k:}$ and $\mathbf{C}_{\mathbf{Y}k:}$ in order to get resampling of $\mathbf{C}_{\mathbf{X}k:}$ and $\mathbf{C}_{\mathbf{Y}k:}$. Specifically, given a random permutation matrix $\mathbf{P}$, $\mathbf{P}\tilde{\mathbf{D}}^{\mathbf{C}_{\mathbf{X}}}_{:,k:}$ and $\tilde{\mathbf{D}}^{\mathbf{C}_{\mathbf{Y}}}_{:,k:}$ together serve as $N$ i.i.d. resamplings from the joint distribution of $\mathbf{C}_{\mathbf{X}k:}$ and $\mathbf{C}_{\mathbf{Y}k:}$. Further, the statistics in Eq. 4 only depends on the $k$-th to $K$-th CCA scores between $\mathbf{X}$ and $\mathbf{Y}$, which can be equivalently calculated by the first to $(K - k)$-th CCA scores between $\mathbf{C}_{\mathbf{X}k:}$ and $\mathbf{C}_{\mathbf{Y}k:}$, formally captured by the following Lemma 1.

**Lemma 1** (Alternative Way to Calculate Statistic in Eq. 4)**.** *Let the CCA score between* $\mathbf{C}_{\mathbf{X}k:}$ *and* $\mathbf{C}_{\mathbf{Y}k:}$ *be* $\{\hat{r}_i\}_1^{K-k}$*. The statistic defined in Eq. 4 can also be formulated as:*

$$\lambda_k = -\left(N - \frac{P + Q + 3}{2}\right)\ln(\Pi_{i=1}^{K-k}(1 - \hat{r}_i^2)). \quad (6)$$

By Lemma 1, we know that the test statistics only depends on $\mathbf{C}_{\mathbf{X}k:}$ and $\mathbf{C}_{\mathbf{Y}k:}$. Further, $\mathbf{C}_{\mathbf{X}k:}$ and $\mathbf{C}_{\mathbf{Y}k:}$ can be resampled by permutations. Taking these two into consideration, we can make use of permutation to estimate the empirical CDF of the null distribution, and thus correctly calculate the p-value. Below we give a detailed description of the procedure to do the permutation and consequently calculate the p-value. Given $\boldsymbol{A}$ and $\boldsymbol{B}$, we have the observed data matrix of the two canonical variables as $\tilde{\boldsymbol{D}}^{\mathbf{C}_{\mathbf{X}}} = \tilde{\boldsymbol{D}}^{\mathbf{X}} \boldsymbol{A}$ and $\tilde{\boldsymbol{D}}^{\mathbf{C}_{\mathbf{Y}}} = \tilde{\boldsymbol{D}}^{\mathbf{Y}} \boldsymbol{B}$ (where $\tilde{\boldsymbol{D}}^{\mathbf{C}_{\mathbf{X}}}, \tilde{\boldsymbol{D}}^{\mathbf{C}_{\mathbf{Y}}} \in \mathbb{R}^{N \times K}$). For each random $N \times N$ permutation matrix $\boldsymbol{P}$, we use $\boldsymbol{P}\tilde{\boldsymbol{D}}^{\mathbf{C}_{\mathbf{X}}}_{:,k:}$ and $\tilde{\boldsymbol{D}}^{\mathbf{C}_{\mathbf{Y}}}_{:,k:}$ to calculate the test statistics under permutation $\boldsymbol{P}$ as $\lambda_k^P$ following Eq. 6, and the p-value is obtained as:

$$p_k = \mathbb{E} \, \mathbf{1}_{[\lambda_k^P \geq \lambda_k]}, \tag{7}$$

where the expectation is taken over random permutations.

### 3.2. Mixed Case - in the Presence of Discretization

Here we discuss the case where some columns of the data matrices $\tilde{\boldsymbol{D}}^{\mathbf{X}}$ and $\tilde{\boldsymbol{D}}^{\mathbf{Y}}$ are discretized. Under such a scenario, one can still estimate $\hat{\Sigma}_{\mathbf{X}}, \hat{\Sigma}_{\mathbf{X},\mathbf{Y}}$, and $\hat{\Sigma}_{\mathbf{Y}}$ by maximizing likelihood, which will be detailed in Section 3.3. After that, $\boldsymbol{A}$ and $\boldsymbol{B}$ can still be estimated following Eq. 5, and the exchangeability between $\mathbf{C}_{\mathbf{X}k:}$ and $\mathbf{C}_{\mathbf{Y}k:}$ still holds.

However, to get the resampling of $\mathbf{C}_{\mathbf{X}k:}$ and $\mathbf{C}_{\mathbf{Y}k:}$ by permutation, one has to apply linear transformation $\boldsymbol{A}$ and $\boldsymbol{B}$ to get $\tilde{\boldsymbol{D}}^{\mathbf{C}_{\mathbf{X}}} = \tilde{\boldsymbol{D}}^{\mathbf{X}} \boldsymbol{A}$ and $\tilde{\boldsymbol{D}}^{\mathbf{C}_{\mathbf{Y}}} = \tilde{\boldsymbol{D}}^{\mathbf{Y}} \boldsymbol{B}$, respectively. In the all continuous case, it is straightforward, but in the presence of discretization, it makes no sense to apply a linear transformation $\boldsymbol{A}$ to $\tilde{\boldsymbol{D}}^{\mathbf{X}}$, when some columns of $\tilde{\boldsymbol{D}}^{\mathbf{X}}$ are just ordinal values. As a consequence, we cannot make use of Theorem 4 to get a resampling of $\mathbf{C}_{\mathbf{X}k:}$ and $\mathbf{C}_{\mathbf{Y}k:}$ to calculate the statistic $\lambda_k$ and estimate the p-value anymore.

Fortunately, it can be shown that to calculate $\lambda_k^P$, one does not have to really get the exact resampling from $\mathbf{C}_{\mathbf{X}k:}$ and $\mathbf{C}_{\mathbf{Y}k:}$. Instead, for each random permutation $\boldsymbol{P}$, we can get a consistent estimation of $\{\hat{r}_i\}_1^{K-k}$ and consequently calculate $\lambda_k^P$. This is formalized by the following Theorem 5.

**Theorem 5** (Consistent Estimation of $\{\hat{r}_i\}_1^{K-k}$ under Permutation $\boldsymbol{P}$). *Under permutation $\boldsymbol{P}$, the empirical CCA scores between $\mathbf{C}_{\mathbf{X}k:}$ and $\mathbf{C}_{\mathbf{Y}k:}$, i.e., $\{\hat{r}_i\}_1^{K-k}$, are the singular values of $\hat{\Sigma}_{\mathbf{C}_{\mathbf{X}k:}}^{-\frac{1}{2}} \hat{\Sigma}_{\mathbf{C}_{\mathbf{X}k:},\mathbf{C}_{\mathbf{Y}k:}} \hat{\Sigma}_{\mathbf{C}_{\mathbf{Y}k:}}^{-\frac{1}{2}}$, which can be consistently estimated by:*

$$((\boldsymbol{A}^T \hat{\Sigma}_{\mathbf{X}} \boldsymbol{A})_{k:,k:})^{-\frac{1}{2}} ((\boldsymbol{A}^T \frac{\boldsymbol{D}^{\mathbf{X}^T} \boldsymbol{P}^T \boldsymbol{D}^{\mathbf{Y}}}{N-1} \boldsymbol{B})_{k:,k:}) \\ ((\boldsymbol{B}^T \hat{\Sigma}_{\mathbf{Y}} \boldsymbol{B})_{k:,k:})^{-\frac{1}{2}}, \tag{8}$$

*where $\frac{\boldsymbol{D}^{\mathbf{X}^T} \boldsymbol{P}^T \boldsymbol{D}^{\mathbf{Y}}}{N-1}$ can be consistently estimated by using $\tilde{\boldsymbol{D}}^{\mathbf{X}}$ and $\boldsymbol{P}^T \tilde{\boldsymbol{D}}^{\mathbf{Y}}$ and assuming unit variance of variables.*

**Remark 1** (Remark on Theorem 5). Theorem 5 implies that we can consistently estimate $\lambda_k^P$ by making use of randomly permuted data $\tilde{\boldsymbol{D}}^{\mathbf{X}}$ and $\boldsymbol{P}^T \tilde{\boldsymbol{D}}^{\mathbf{Y}}$. Note that although here the transpose of permutation applies to $\tilde{\boldsymbol{D}}^{\mathbf{Y}}$, the correctness of the process still relies on the exchangeability between $\mathbf{C}_{\mathbf{X}k:}$ and $\mathbf{C}_{\mathbf{Y}k:}$, and does not need the exchangeability between $\mathbf{X}$ and $\mathbf{Y}$. In words, doing permutation on $\tilde{\boldsymbol{D}}^{\mathbf{X}} \boldsymbol{A}$ will meet the problem of applying linear transformation to data that might contain ordinal values, and Theorem 5 provides a way to bypass the problem by permuting $\tilde{\boldsymbol{D}}^{\mathbf{Y}}$ instead.

Till now, the remaining problem is how to consistently estimate cross-covariance matrices in the presence of discretization, and it will be detailed in what follows.

### 3.3. Correlation Estimation with Discretization

Assume that we concern the rank of $\Sigma_{\mathbf{X},\mathbf{Y}}$, where some of the variables are discretized and $\mathbf{X}$ and $\mathbf{Y}$ are not necessarily disjoint. As mentioned, for those variables that we only have discretized observations, their variance can never be determined. Further, the rank of a cross-covariance matrix is equivalent to the rank of the corresponding cross-correlation matrix. Without loss of generality, we can assume all variables to have unit variance and zero mean. Thus, we sometimes use correlation and covariance interchangeably. The remaining crucial step is to estimate the correlation matrix for $\mathbf{V} = \mathbf{X} \cup \mathbf{Y}$, i.e., $\hat{\boldsymbol{R}}$, by data $\tilde{\boldsymbol{D}} \in \mathbb{R}^{N \times |\boldsymbol{V}|}$. As some elements of $\mathbf{V}$ are discrete, we use $\mathbb{C}_{\mathbf{V}}$ and $\mathbb{D}_{\mathbf{V}}$ to denote the index set of continuous variables and discrete variables in $\mathbf{V}$ respectively.

We first introduce the overall objective function for correlation estimation as follows.

$$\hat{\boldsymbol{R}} = \arg\min_{\boldsymbol{R} \in \mathbb{R}^{M \times M}} \mathcal{L}(\tilde{\boldsymbol{D}}, \boldsymbol{R}), \tag{9}$$

$$\mathcal{L}(\tilde{\boldsymbol{D}}, \boldsymbol{R}) = - \sum_{1 \leq i < j \leq M} \log p_{ij}(\tilde{\boldsymbol{D}}_{:,ij}; \boldsymbol{R}_{i,j}), \tag{10}$$

where the optimization objective is minimizing pair-wise negative log-likelihood, also referred to as pseudo likelihood, instead of the real joint log-likelihood over all the observed variables (Dong et al., 2024b). The reason lies in that optimizing over the joint log-likelihood is very computationally expensive and the pseudo likelihood is tractable while also serves as a consistent estimator (Besag, 1974; Gourieroux et al., 1984; Gouriéroux et al., 2017; Fan et al., 2017).

Next, we specify the pair-wise log-likelihood in three scenarios - between two continuous variables, between a continuous and a discrete, and between two discrete variables.

#### (i) Likelihood for Two Continuous Variables

If both $i, j \in \mathbb{C}_{\mathbf{V}}$, the log-likelihood function $\log p_{ij}(\tilde{\boldsymbol{D}}_{:,ij}; \boldsymbol{R}_{i,j})$ is just the joint gaussian pdf

parametrized by $\boldsymbol{R}_{i,j}$ given as follows:

$$(1/2)(\text{tr}\left(\begin{bmatrix} 1, \boldsymbol{R}_{i,j} \\ \boldsymbol{R}_{i,j}, 1 \end{bmatrix}^{-1} \begin{bmatrix} 1, \hat{\boldsymbol{R}}_{i,j} \\ \hat{\boldsymbol{R}}_{i,j}, 1 \end{bmatrix}\right) + \log \det \begin{bmatrix} 1, \boldsymbol{R}_{i,j} \\ \boldsymbol{R}_{i,j}, 1 \end{bmatrix}), \tag{11}$$

where $\hat{\boldsymbol{R}}_{i,j}$ is the empirical correlation matrix that can be directly calculated from data $\tilde{\boldsymbol{D}}_{:,ij}$.

**(ii) Likelihood for a Continuous and a Discrete Variable**

If $i \in \mathbb{C}_{\mathbf{V}}$ and $j \in \mathbb{D}_{\mathbf{V}}$, then the log-likelihood (also known as polyserial correlation estimation (Olsson et al., 1982)) $\log p_{ij}(\tilde{\boldsymbol{D}}_{:,ij}; \boldsymbol{R}_{i,j})$ can be factorized as follows.

$$\frac{1}{N} \sum_{k=1}^{N} \log p(\mathsf{V}_i = \tilde{\boldsymbol{D}}_{k,i}) p(\mathsf{V}_j = \tilde{\boldsymbol{D}}_{k,j} | \mathsf{V}_i = \tilde{\boldsymbol{D}}_{k,i}, \boldsymbol{R}_{i,j}), \tag{12}$$

where $p(\mathsf{V}_i = \tilde{\boldsymbol{D}}_{k,i})$ is a standard gaussian pdf. For a specific value of $\tilde{\boldsymbol{D}}_{k,j}$, say, $t$, we have that:

$$\begin{aligned} &p(\mathsf{V}_j = \tilde{\boldsymbol{D}}_{k,j} | \mathsf{V}_i = \tilde{\boldsymbol{D}}_{k,i}, \boldsymbol{R}_{i,j}) \\ &= p(T_t^j < \mathsf{V}_j \leq T_{t+1}^j | \mathsf{V}_i = \tilde{\boldsymbol{D}}_{k,i}, \boldsymbol{R}_{i,j}) \\ &= \Phi(\frac{T_{t+1}^j - \boldsymbol{R}_{i,j}\tilde{\boldsymbol{D}}_{k,i}}{(1 - \boldsymbol{R}_{i,j}^2)^{1/2}}) - \Phi(\frac{T_t^j - \boldsymbol{R}_{i,j}\tilde{\boldsymbol{D}}_{k,i}}{(1 - \boldsymbol{R}_{i,j}^2)^{1/2}}), \end{aligned} \tag{13}$$

where $\Phi$ is the standard gaussian cdf. We note that the thresholds $T$ are unknown, thus it could be taken as free parameters during optimization. In practice, it is more efficient to estimate the thresholds first by using inverse gaussian cdf:

$$\hat{T}_{t+1}^j = \Phi^{-1}(\frac{\sum_{k=1}^{N} \mathbf{1}_{[\tilde{\boldsymbol{D}}_{k,j} \leq t]}}{N}). \tag{14}$$

**(iii) Likelihood for Two Discrete Variables**

If both $i, j \in \mathbb{D}_{\mathbf{V}}$, then the log-likelihood (also known as polychoric correlation estimation (Olsson, 1979; Jöreskog, 1994)) $\log p_{ij}(\tilde{\boldsymbol{D}}_{:,ij}; \boldsymbol{R}_{i,j})$ is as follows.

$$\begin{aligned} \frac{1}{N} \sum_{k=1}^{N} \log(&\Phi_2(T_{\tilde{\boldsymbol{D}}_{k,i}+1}^i, T_{\tilde{\boldsymbol{D}}_{k,j}+1}^j; \boldsymbol{R}_{i,j}) \\ &+ \Phi_2(T_{\tilde{\boldsymbol{D}}_{k,i}}^i, T_{\tilde{\boldsymbol{D}}_{k,j}}^j; \boldsymbol{R}_{i,j}) \\ &- \Phi_2(T_{\tilde{\boldsymbol{D}}_{k,i}+1}^i, T_{\tilde{\boldsymbol{D}}_{k,j}}^j; \boldsymbol{R}_{i,j}) \\ &- \Phi_2(T_{\tilde{\boldsymbol{D}}_{k,i}}^i, T_{\tilde{\boldsymbol{D}}_{k,j}+1}^j; \boldsymbol{R}_{i,j})), \end{aligned} \tag{15}$$

where $\Phi_2(.,.,r)$ is the joint cdf of two standard gaussian variables with correlation $r$ and the thresholds for each variable can also be estimated by using Eq. 14.

---

**Algorithm 1** Mixed data Permutation-based Rank Test

1: **Input:** Sample $\tilde{D}^{\mathbf{X}}$, $\tilde{D}^{\mathbf{Y}}$, indexes of discretized columns, null hypothesis $\mathcal{H}_0^k : \text{rank}(\Sigma_{\mathbf{X},\mathbf{Y}}) \leq k$, and significant level $\alpha$;
2: **Output:** True (fail to reject $\mathcal{H}_0^k$) or False (reject $\mathcal{H}_0^k$);
3: $P = |\mathbf{X}|$, $Q = |\mathbf{Y}|$, and $K = \min(P, Q)$
4: Get $\hat{\Sigma}_{\mathbf{X}}, \hat{\Sigma}_{\mathbf{X},\mathbf{Y}}$, and $\hat{\Sigma}_{\mathbf{Y}}$ as submatrices of $\hat{\boldsymbol{R}}$ by Eq. 17 (unit variance assumed)
5: Calculate $\boldsymbol{A}$ and $\boldsymbol{B}$ following Eq. 5.
6: Let $\boldsymbol{P} = \boldsymbol{I}$ (no permutation), calculate $\{\hat{r}_i\}_1^{K-k}$ following Eq. 8 and then the statistic $\lambda_k$ following Eq. 6
7: **for** each random permutation $\boldsymbol{P}$ **do**
8:     Calculate $\{\hat{r}_i\}_1^{K-k}$ under $\boldsymbol{P}$ following Eq. 8 and then the statistic under $\boldsymbol{P}$, i.e., $\lambda_k^P$, following Eq. 4
9: **end for**
10: Calculate p-value $p_k$ by Eq. 7
11: **return** $p_k \geq \alpha$

---

### 3.4. Parameterization Trick for Rank Test

We note that the optimization problem in Eq. 9 does not constrain the space to be a pseudo-correlation matrix - a matrix that is PSD with unit diagonal elements. If we only care about the maximum likelihood estimator, the pseudo-correlation requirement might be unnecessary. However, as we rely on SVD for CCA and rank test, the requirement of being pseudo-correlation matrix is crucial. A classical way to solve this problem is by projected gradient descent: projecting the current solution to the space of pseudo-correlation matrices after each step of gradient descent. Yet, in practice we found this solution less effective, as the projection cannot be analytically solved and requires an additional optimization step.

To this end, we directly parameterize the space of pseudo-correlation matrices in a geometric way following (Rousseeuw & Molenberghs, 1993), given as follows.

$$\begin{aligned} \boldsymbol{R} &= \boldsymbol{U}^T \boldsymbol{U}, \\ \boldsymbol{U}_{j,i} &= \begin{cases} \cos\boldsymbol{\theta}_{i-j+1,i}\Pi_{k=1}^{i-j}\sin\boldsymbol{\theta}_{k,i}, & j \leq i \\ 0, & j > i \end{cases}, \\ &\text{s.t., } \boldsymbol{\theta}_{i,i} = 0, \forall i. \end{aligned} \tag{16}$$

Therefore, we have an alternative way to parameterize the correlation matrix, which gives rise to the following new formulation of our objective function (instead of Eq. 9):

$$\hat{\boldsymbol{R}} = \arg\min_{\boldsymbol{\theta}} \mathcal{L}(\tilde{\boldsymbol{D}}, \boldsymbol{R}). \tag{17}$$

We summarize the overall testing procedure of our proposed MPRT in Algorithm 1.

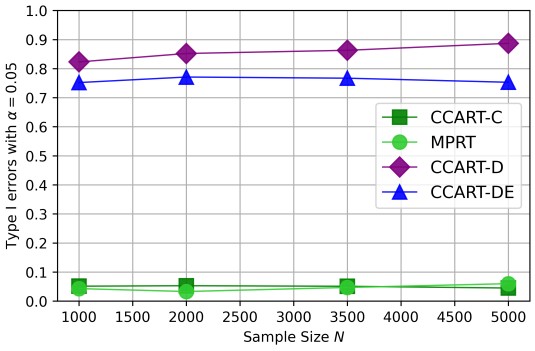

(a) The probability of Type I errors with $\alpha = 0.05$.

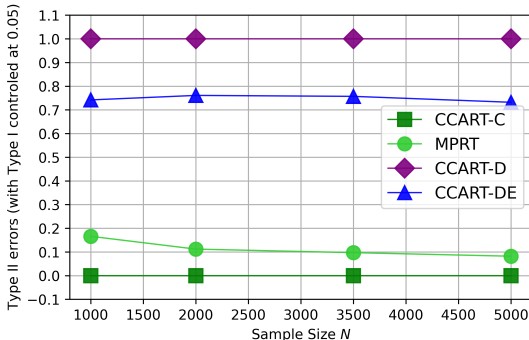

(b) Type II errors (effective Type I controlled at $0.05$).

*Figure 2.* The probability of Type I and Type II errors with **mixed data**, by different rank test methods, under different sample sizes.

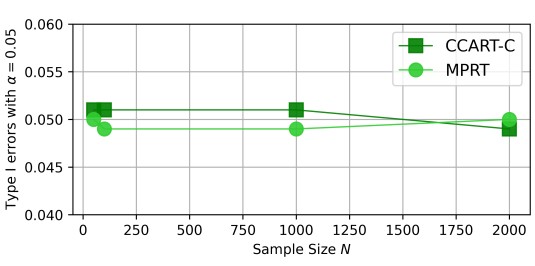

(a) The probability of Type I errors with $\alpha = 0.05$.

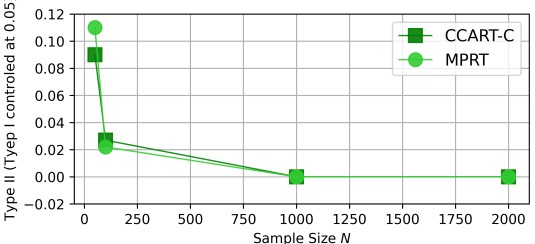

(b) Type II errors (effective Type I controlled at $0.05$).

*Figure 3.* The probability of Type I and Type II errors with **continuous data**, by different rank test methods, under different sample sizes.

## 4. Experiments

### 4.1. Experimental Setting

To empirically validate the proposed Mixed data Permutation-based Rank Test (MPRT), we apply our method to synthetic data and compare it with the following methods. (i) CCART-C: CCA-based Rank Test (Anderson, 1984) that use the original continuous observation as input; as it has access to the original observations, its performance is taken as the best possible performance that we can achieve. (ii) CCART-D: CCA-based Rank Test with Discrete data; it directly takes the ordinal values as input. (iii) CCART-DE: CCA-based Rank Test with Discrete data Estimating covariance; it takes the estimated correlation matrix as input (following Eq. 17).

We consider two scenarios: mixed data scenario where data are partially discretized, and all continuous scenario where all the original observations are available. The first scenario is to illustrate how well can we handle discretization while the second is to show that our method can serve as a general rank test method as we also work well when there is no discretization. In terms of performance, we concern both Type I errors and Type II errors. Specifically, we expect a good test can properly control the Type I errors given a significance level $\alpha$, while the Type II errors should be as

small as possible. We consider different sample sizes, and for each comparison, we consider 3000 random trials. For MPRT, we randomly generated 200 permutations to calculate the p-value. The ground truth covariance matrices are randomly generated. For the mixed scenario, we uniformly generate two thresholds from $[-1.5, 1.5]$ for each variable that should be discretized, and use the thresholds together with $-\infty$ and $\infty$ to discretize the continuous observations into three categories $\{1, 2, 3\}$.

We also apply the proposed MPRT method with mixed data to the classical causal discovery method PC algorithm (Spirtes et al., 2000) and see whether our test method can better test CI relations compared to the classical Fisher-Z CI test (Fisher et al., 1921), in the presence of discretization. Fisher-Z is only compared by the result of PC and cannot be not compared in the previous setting, as linear CI relations can only correspond to a part of the rank information. Finally, we employ a real-life dataset to illustrate the applicability of the proposed method in real-life scenarios.

### 4.2. Analysis on Type I and Type II Errors under Different Sample Sizes

In this section we analyze the performance of each method in terms of Type I and Type II errors under different sample sizes. For the mixed data scenario, the result is shown in

*Table 1.* F1 score and SHD of the PC algorithm, with different CI test methods (↑ the bigger the better while ↓ the smaller the better).

| CI test method | F1 score for skeleton ↑ | | | SHD for skeleton ↓ | | |
|---|---|---|---|---|---|---|
| | $N = 500$ | $N = 1000$ | $N = 2000$ | $N = 500$ | $N = 1000$ | $N = 2000$ |
| **MPRT** | **0.84** | **0.9** | **0.96** | **0.80** | **0.60** | **0.20** |
| Fisher-Z | 0.81 | 0.80 | 0.78 | 1.20 | 1.20 | 1.40 |
| KCI | 0.81 | 0.88 | 0.86 | 1.00 | 0.80 | 0.93 |
| CCART-D | 0.75 | 0.79 | 0.77 | 1.60 | 1.60 | 1.80 |
| CCART-DE | 0.80 | 0.85 | 0.83 | 1.40 | 1.30 | 1.60 |

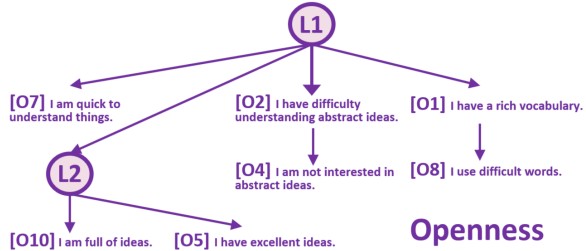

(a) Discovered personality substructure for Openness.

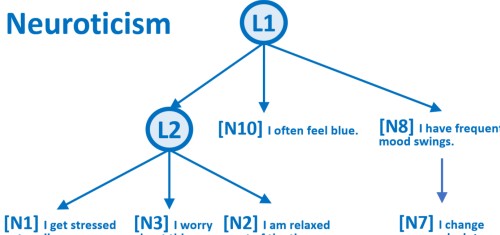

(b) Discovered substructure for Neuroticism.

*Figure 4.* Application of MPRT in causal discovery using real-life Big Five human personality data.

Figure 2. Specifically, one can see that both our proposed MPRT and CCART-C can properly control the Type I errors as the Type I errors of them are both very close to the significance level $\alpha = 0.05$; in contrast, CCART-D and CCART-DE totally failed to control the Type I errors. As for Type II errors, it can be found that the Type II errors of MPRT are quite small, and decreases with the increase of sample size $N$, while CCART-D and CCART-DE cannot benefit from the increase in sample size. We note that it is very natural that MPRT cannot beat CCART-C as CCART-C takes the original continuous observation as input while MPRT takes mixed data as input. We show the performance of CCART-C just in order to show the minimal possible Type II errors that one can achieve in the presence of discretization.

We also show the performance when both CCART-C and MPRT have access to the original continuous observations, as in Figure 3. Specifically, both methods properly control the Type I errors as in the subfigure 3 (a). For the Type II errors, the performance of CCART-C and MPRT is almost the same. This is as expected, as in this scenario both methods use exactly the same test statistics except that CCART-C uses the analytically derived null distribution to get the p-value while MPRT uses the empirical CDF to calculate the p-value; the two results are expected to be exactly the same asymptotically.

Taking the performance under these two scenarios together into consideration it can be argued that MPRT is a very general and valid rank test as it can handle all continous data, partially discretized data, and all discretized data and the Type I are properly controlled while the power is also good.

### 4.3. Application in Causal Discovery

In this section we validate our test using the PC algorithm (Spirtes et al., 2000). Specifically, we consider linear causal models with gaussian noises $V_i = \sum_{V_j \in \text{Pa}(V_i)} a_{ij} V_j + \varepsilon_{V_i}$, where the edge coefficients and the variance of the noises are randomly generated. We consider the scenario where data are partially discretized and compare MPRT with Fisher-Z to see which one works better with PC. We employ F1 score $F1 = \frac{2*\text{Recall}*\text{Precision}}{\text{Recall}+\text{Precision}}$ for skeleton (the bigger the better) and Structural Hamming Distance (SHD) for skeleton (the smaller the better) to evaluate the performance. As shown in Table 1, MPRT achieves the best performance in terms of both F1 and SHD, under all sample sizes. This validates the claim that MPRT can serve as a powerful CI test for causal discovery in the presence of discretization.

### 4.4. Real-world Causal Discovery Application

In this section, we further validate our proposed MPRT method using a real-world Big Five Personality dataset https://openpsychometrics.org/. It consists of 50 personality indicators and close to 20,000 data points. Each Big Five personality dimension, namely, Openness, Conscientiousness, Extraversion, Agreeableness, and Neuroticism (O-C-E-A-N), are designed to be measured with their own 10 indicators and the values of each variable are ordinal: Disagree, slightly disagree, Neutral, Slightly agree, and Agree. We employ RLCD (Dong et al., 2024a), a recently proposed rank based causal discovery method with our MPRT method. We choose 7 items from openness and 6 items from neuroticism to verify our method.

The results are shown in Figure 4. Specifically, for open-

ness we discovered two latent variables. L2 corresponds to whether a person has a lot of ideas while L1 corresponds to the general concept of openness. As for neuroticism, we also discovered two latent variables. L1 relates more to one's emotions while L2 relates to one's stress level. In contrast, if we directly use the ordinal values to do the rank test, i.e., using CCART-D, all the p-values tend to be very small, and thus we have to use very small significance level (around 1e-10) in order to have some structures discovered; yet using such an extremely small alpha value will induce a lot of Type II errors. This result illustrates the superiority of using MPRT in the presence of discretizations in real-life scenarios, and again empirically validate the proposed method.

### 4.5. Discussion about Unit Variance Assumption in Correlation Estimation and Non-Gaussianity

In Section 3.3, we assume that the underlying continuous variables have unit variance and zero mean. Violation of this assumption, i.e., shift and rescaling of variables, does not affect the validity of our method. This is because we care about the rank of the cross-covariance matrix, which is equal to the rank of the cross-correlation matrix; the latter is clearly invariant to shift or rescaling of either some or all variables. Thus, in Section 3.3 we assume all variables are standardized just for simplicity of notation.

If we assume that the underlying continuous variables follow a linear SCM, but the joint distribution are not necessarily gaussian anymore, the proposed method can still work, as long as the parametric form is given: we only need to modify the likelihood function in Section 3.3 according to the corresponding parametric form for correlation estimation. As a comparison, traditional CCA-based rank tests must assume normality to infer the null distribution. On the other hand, if the parametric form is not given, which means we do not have any information about the shape of the distribution, it may be very hard to consistently recover the thresholds and the underlying correlation, due to insufficient information.

## 5. Conclusion

In this paper, we propose a novel permutation-based rank test that works in the presence of discretization. It is rather general as it can accommodate fully continuous data, partially discretized data, or fully discretized data as input. Extensive experiments empirically validate our method.

## Impact Statement

This paper presents work whose goal is to advance the field of Machine Learning. There are many potential societal consequences of our work, none which we feel must be specifically highlighted here.

## Acknowledgment

We would like to acknowledge the support from NSF Award No. 2229881, AI Institute for Societal Decision Making (AI-SDM), the National Institutes of Health (NIH) under Contract R01HL159805, and grants from Quris AI, Florin Court Capital, and MBZUAI-WIS Joint Program. IN acknowledges the support of the Natural Sciences and Engineering Research Council of Canada (NSERC) Postgraduate Scholarships – Doctoral program.

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

# A. Proofs

## A.1. Proof of Theorem 4

**Theorem 4** (Exchangeability of $\mathbf{C}_{\mathbf{X}\,k:}$ and $\mathbf{C}_{\mathbf{Y}\,k:}$). *Given a set of variables $\mathbf{V}$ that are jointly gaussian, under null hypothesis $\mathcal{H}_0^k : rank(\Sigma_{\mathbf{X},\mathbf{Y}}) \leq k$, where $\mathbf{X}, \mathbf{Y} \subseteq \mathbf{V}$, random vectors $\mathbf{C}_{\mathbf{X}\,k:}$ and $\mathbf{C}_{\mathbf{Y}\,k:}$ are asymptotically independent with each other.*

*Proof of Theorem 4.* First, $\hat{\Sigma}_{\mathbf{X}}$, $\hat{\Sigma}_{\mathbf{Y}}$, and $\hat{\Sigma}_{\mathbf{X},\mathbf{Y}}$ by pseudo-likelihood, converge in probability to $\Sigma_{\mathbf{X}}$, $\Sigma_{\mathbf{Y}}$, and $\Sigma_{\mathbf{X},\mathbf{Y}}$, respectively (Besag, 1974; Gourieroux et al., 1984; Gouriéroux et al., 2017; Fan et al., 2017).

Plus, as we need to apply the continuous mapping theorem, we show the continuity and uniqueness of SVD in what follows. SVD is not continuous only when the input matrix has repeated singular values. Specifically, if a matrix $A$ has distinct singular values, then SVD is continuous in the neighborhood of $A$, and unique only up to sign flip (chapter 2 section 5.3 of (Kato, 2013)). Thus, to make use of the continuous mapping theorem, we assume that $\Sigma_{\mathbf{X}}^{-\frac{1}{2}}\Sigma_{\mathbf{X},\mathbf{Y}}\Sigma_{\mathbf{Y}}^{-\frac{1}{2}}$ does not have repeated singular values (the set of matrices with repeated singular values has Lebesgue measure zero (Lemma 1.4.2 in (Kunisky), also in (Bochnak et al., 2013)).). To further eliminate the sign indeterminacy, we can just follow scikit-learn to impose the largest coefficient of each column in $U$ in absolute value is positive (svd flip in scikit-learn).

Given $(\hat{\Sigma}_{\mathbf{X}}, \hat{\Sigma}_{\mathbf{Y}}, \hat{\Sigma}_{\mathbf{X},\mathbf{Y}}) \xrightarrow{p} (\Sigma_{\mathbf{X}}, \Sigma_{\mathbf{Y}}, \Sigma_{\mathbf{X},\mathbf{Y}})$, we aim to show the desired asymptotic independence. Specifically we want to show (i) $\mathbf{C}_{\mathbf{X}\,k:} \xrightarrow{p} \mathbf{C}_{\mathbf{X}\,k:}^*$ and $\mathbf{C}_{\mathbf{Y}\,k:} \xrightarrow{p} \mathbf{C}_{\mathbf{Y}\,k:}^*$, and (ii) $\mathbf{C}_{\mathbf{X}\,k:}^*, \mathbf{C}_{\mathbf{Y}\,k:}^*$ are independent under the null hypo. Here $\mathbf{C}_{\mathbf{X}} = A^T\mathbf{X}, \mathbf{C}_{\mathbf{Y}} = B^T\mathbf{Y}, \mathbf{C}_{\mathbf{X}}^* = A^{*T}\mathbf{X}$, and $\mathbf{C}_{\mathbf{Y}}^* = B^{*T}\mathbf{Y}$, where $(A, B)$ and $(A^*, B^*)$ are produced by SVD using estimated covariance and population one respectively as follows.

$$USV = \hat{\Sigma}_{\mathbf{X}}^{-\frac{1}{2}}\hat{\Sigma}_{\mathbf{X},\mathbf{Y}}\hat{\Sigma}_{\mathbf{Y}}^{-\frac{1}{2}}, A = \hat{\Sigma}_{\mathbf{X}}^{-\frac{1}{2}T}U, B = \hat{\Sigma}_{\mathbf{Y}}^{-\frac{1}{2}T}V^T, \ U^*S^*V^* = \Sigma_{\mathbf{X}}^{-\frac{1}{2}}\Sigma_{\mathbf{X},\mathbf{Y}}\Sigma_{\mathbf{Y}}^{-\frac{1}{2}}, A^* = \Sigma_{\mathbf{X}}^{-\frac{1}{2}T}U^*, B^* = \Sigma_{\mathbf{Y}}^{-\frac{1}{2}T}V^{*T}.$$

For (i): By continuous mapping theorem, under the assumption of no repeated singular values, we have $U \xrightarrow{p} U^*$. As $\Sigma_{\mathbf{X}}$ is positive definite, the matrix inverse square root is continuous and thus $\hat{\Sigma}_{\mathbf{X}}^{-\frac{1}{2}T} \xrightarrow{p} \Sigma_{\mathbf{X}}^{-\frac{1}{2}T}$. Given $(U, \hat{\Sigma}_{\mathbf{X}}^{-\frac{1}{2}T}) \xrightarrow{p} (U^*, \Sigma_{\mathbf{X}}^{-\frac{1}{2}T})$, we have $\hat{\Sigma}_{\mathbf{X}}^{-\frac{1}{2}T}U = A \xrightarrow{p} A^* = \Sigma_{\mathbf{X}}^{-\frac{1}{2}T}U^*$. Similarly, we have $B \xrightarrow{p} B^*$. Thus

$$((A^T - A^{*T})\mathbf{X}, (B^T - B^{*T})\mathbf{Y}) \xrightarrow{p} 0 \Rightarrow (((A^T - A^{*T})\mathbf{X})_{k:}, ((B^T - B^{*T})\mathbf{Y})_{k:}) \xrightarrow{p} 0 \Rightarrow (\mathbf{C}_{\mathbf{X}\,k:}, \mathbf{C}_{\mathbf{Y}\,k:}) \xrightarrow{p} (\mathbf{C}_{\mathbf{X}\,k:}^*, \mathbf{C}_{\mathbf{Y}\,k:}^*).$$

For (ii): Under the null hypo, the cross-covariance between $\mathbf{C}_{\mathbf{X}\,k:}^*$ and $\mathbf{C}_{\mathbf{Y}\,k:}^*$ are all zeros. As $\mathbf{C}_{\mathbf{X}}\_k:^*, \mathbf{C}_{\mathbf{Y}\,k:}^*$ are jointly gaussian (linear mixing of $\mathbf{X}, \mathbf{Y}$), zero cross-covariance implies independence.

$\square$

## A.2. Proof of Lemma 1

**Lemma 1** (Alternative Way to Calculate Statistic in Eq. 4). *Let the CCA score between $\mathbf{C}_{\mathbf{X}\,k:}$ and $\mathbf{C}_{\mathbf{Y}\,k:}$ be $\{\hat{r}_i\}_1^{K-k}$. The statistic defined in Eq. 4 can also be formulated as:*

$$\lambda_k = -\left(N - \frac{P+Q+3}{2}\right)\ln(\Pi_{i=1}^{K-k}(1 - \hat{r}_i^2)). \tag{6}$$

*Proof of Lemma 1.* The CCA scores between $\mathbf{C}_{\mathbf{X}\,k:}$ and $\mathbf{C}_{\mathbf{Y}\,k:}$ are just the diagonal entries of their cross-covariance matrix, which corresponds to the $k$ to $K$ CCA scores between $\mathbf{X}$ and $\mathbf{Y}$. Thus we have $\hat{r}_i = r_{i+k}$ for $i = \{1, ..., K-k\}$, and thus $\lambda_k = -(N - \frac{P+Q+3}{2})\ln(\Pi_{i=k+1}^K(1 - r_i^2))$. $\square$

## A.3. Proof of Theorem 5

**Theorem 5** (Consistent Estimation of $\{\hat{r}_i\}_1^{K-k}$ under Permutation $\boldsymbol{P}$). *Under permutation $\boldsymbol{P}$, the empirical CCA scores between $\mathbf{C}_{\mathbf{X}\,k:}$ and $\mathbf{C}_{\mathbf{Y}\,k:}$, i.e., $\{\hat{r}_i\}_1^{K-k}$, are the singular values of $\hat{\Sigma}_{\mathbf{C}_{\mathbf{X}k:}}^{-\frac{1}{2}}\hat{\Sigma}_{\mathbf{C}_{\mathbf{X}k:},\mathbf{C}_{\mathbf{Y}k:}}\hat{\Sigma}_{\mathbf{C}_{\mathbf{Y}k:}}^{-\frac{1}{2}}$, which can be consistently estimated by:*

$$((\boldsymbol{A}^T\hat{\Sigma}_{\mathbf{X}}\boldsymbol{A})_{k:,k:})^{-\frac{1}{2}}((\boldsymbol{A}^T\frac{\boldsymbol{D}^{\mathbf{X}T}\boldsymbol{P}^T\boldsymbol{D}^{\mathbf{Y}}}{N-1}\boldsymbol{B})_{k:,k:})$$
$$((\boldsymbol{B}^T\hat{\Sigma}_{\mathbf{Y}}\boldsymbol{B})_{k:,k:})^{-\frac{1}{2}}, \tag{8}$$

where $\frac{D^{X^T} P^T D^Y}{N-1}$ can be consistently estimated by using $\tilde{D}^X$ and $P^T \tilde{D}^Y$ and assuming unit variance of variables.

*Proof of Theorem 5.* We are interested in $\hat{\Sigma}_{C_{X k:}}^{-\frac{1}{2}} \hat{\Sigma}_{C_{X k:}, C_{Y k:}} \hat{\Sigma}_{C_{Y k:}}^{-\frac{1}{2}}$. Assume that we have access to the original data $D^X$ and $D^Y$. By the exchangeability, for each random $P$, we have $(PD^X A)_{:,k:}$ and $(D^Y B)_{:,k:}$ are the $N$ samples from joint distribution of $C_{X k:}$ and $C_{Y k:}$. Then the $\hat{\Sigma}_{C_{X k:}}^{-\frac{1}{2}}$, $\hat{\Sigma}_{C_{X k:}, C_{Y k:}}$, and $\hat{\Sigma}_{C_{Y k:}}^{-\frac{1}{2}}$ are as follows:

$$\hat{\Sigma}_{C_{X k:}}^{-\frac{1}{2}} = (\frac{((PD^X A)_{:,k:})^T (PD^X A)_{:,k:}}{N-1})^{-\frac{1}{2}}, \tag{18}$$

$$= (\frac{((PD^X A)^T (PD^X A))_{k:,k:}}{N-1})^{-\frac{1}{2}}, \tag{19}$$

$$= (\frac{(A^T D^{X^T} D^X A)_{k:,k:}}{N-1})^{-\frac{1}{2}}, \tag{20}$$

$$= ((A^T \hat{\Sigma}_X A)_{k:,k:})^{-\frac{1}{2}}. \tag{21}$$

$$\hat{\Sigma}_{C_{Y k:}}^{-\frac{1}{2}} = (\frac{((D^Y B)_{:,k:})^T (D^Y B)_{:,k:}}{N-1})^{-\frac{1}{2}}, \tag{22}$$

$$= (\frac{((D^Y B)^T (D^Y B))_{k:,k:}}{N-1})^{-\frac{1}{2}}, \tag{23}$$

$$= (\frac{(B^T D^{Y^T} D^Y B)_{k:,k:}}{N-1})^{-\frac{1}{2}}, \tag{24}$$

$$= ((B^T \hat{\Sigma}_Y B)_{k:,k:})^{-\frac{1}{2}}. \tag{25}$$

$$\hat{\Sigma}_{C_{X k:}, C_{Y k:}} = \frac{((PD^X A)_{:,k:})^T (D^Y B)_{:,k:}}{N-1}, \tag{26}$$

$$= \frac{((PD^X A)^T D^Y B)_{k:,k:}}{N-1}, \tag{27}$$

$$= (\frac{(A^T D^{X^T} P^T D^Y B)_{k:,k:}}{N-1}), \tag{28}$$

$$= (A^T \frac{D^{X^T} P^T D^Y}{N-1} B)_{k:,k:}. \tag{29}$$

Further, $\tilde{D}^X$ and $P^T \tilde{D}^Y$ can be taken as sampled from the joint distribution of two independent gaussian random vectors. As each of them are marginally gaussian, they are also jointly gaussian. Thus, $\frac{D^{X^T} P^T D^Y}{N-1}$ can be consistently estimated by maximizing likeilhood as in Eq. 17. □

## B. Other Definitions

### B.1. T-separation

The definitions of trek and t-separation are as follows.

**Definition 1** (Treks (Sullivant et al., 2010)). In $\mathcal{G}$, a trek from X to Y is an ordered pair of directed paths $(P_1, P_2)$ where $P_1$ has a sink X, $P_2$ has a sink Y, and both $P_1$ and $P_2$ have the same source Z.

**Definition 2** (T-separation (Sullivant et al., 2010)). Let $\mathbf{A}$, $\mathbf{B}$, $\mathbf{C_A}$, and $\mathbf{C_B}$ be four subsets of $\mathbf{V}_{\mathcal{G}}$ in graph $\mathcal{G}$ (not necessarilly disjoint). $(\mathbf{C_A}, \mathbf{C_B})$ t-separates $\mathbf{A}$ from $\mathbf{B}$ if for every trek $(P_1, P_2)$ from a vertex in $\mathbf{A}$ to a vertex in $\mathbf{B}$, either $P_1$ contains a vertex in $\mathbf{C_A}$ or $P_2$ contains a vertex in $\mathbf{C_B}$.

**Example 1.** In Figure 5, there are multiple treks. For example, $X_4 \leftarrow L_1 \rightarrow X_3$ is a trek between $X_4$ and $X_3$, $X_4 \leftarrow L_1$ is a trek between $X_4$ and $L_1$, and $L_1 \rightarrow X_3$ is a trek between $L_1$ and $X_3$. As for t-separations, we have $\{X_1, X_2\}$ and $\{X_3, X_4\}$ are t-separated by $(\emptyset, \{L_1\})$.

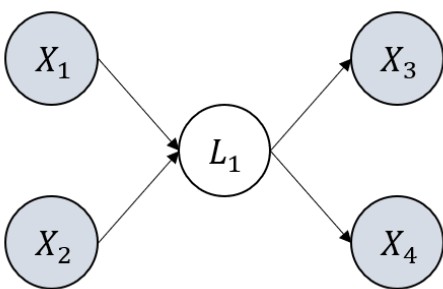

*Figure 5.* An illustrative example to show that rank contains more graphical information than CI. When using CI, we cannot deduce that $\{\mathbf{X}_1, \mathbf{X}_2\}$ and $\{\mathbf{X}_3, \mathbf{X}_4\}$ are d-separated by $\mathbf{L}_1$ as $\mathbf{L}_1$ is latent, while by using rank we can.

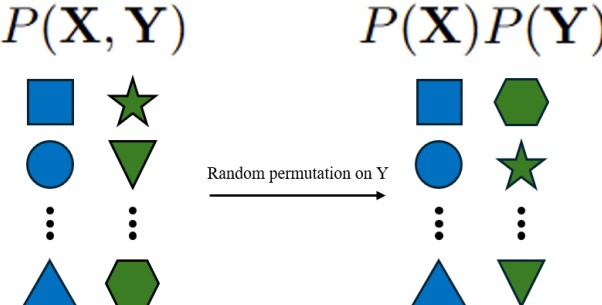

*Figure 6.* An illustration of exchangeability and permutation test. The left figure refer to $N$ i.i.d. samples from $P(\mathbf{X}, \mathbf{Y})$. After random permutation on $\mathbf{Y}$, the permutated data can be considered as random i.i.d. samples from $P(\mathbf{X})$ and $P(\mathbf{Y})$. If the exchangeability holds, i.e., random vectors $\mathbf{X}$ and $\mathbf{Y}$ are independent, then we have $P(\mathbf{X}, \mathbf{Y}) = P(\mathbf{X})P(\mathbf{Y})$, and thus the permuted data can serve as another $N$ i.i.d. samples from $P(\mathbf{X}, \mathbf{Y})$.

## C. Discussion

### C.1. Brief Introduction to Permutation Test

Permutation tests aim to empirically estimate the CDF of the null distribution of a test statistic. The core of such an CDF estimation is the exchangeability, under which we can make use of permuted data to serve as additional samples from the same distribution.

Take Figure 6 as an example. The left figure in Figure 6 refer to $N$ i.i.d. samples from $P(\mathbf{X}, \mathbf{Y})$. After random permutation on $\mathbf{Y}$, the permutated data can be considered as random i.i.d. samples from $P(\mathbf{X})$ and $P(\mathbf{Y})$. If the exchangeability holds under the null hypothesis, i.e., random vectors $\mathbf{X}$ and $\mathbf{Y}$ are independent, then we have $P(\mathbf{X}, \mathbf{Y}) = P(\mathbf{X})P(\mathbf{Y})$, and thus the permuted data can serve as another $N$ i.i.d. samples from $P(\mathbf{X}, \mathbf{Y})$. Now we know how to generate additional $N$ i.i.d. samples. As a test statistic is just a deterministic function of the $N$ i.i.d., samples. For each randomly permuted data, we can calculate the value of the test statistic, and thus all these calculated test statistics can be considered as sampled from the distribution of the test statistic. Given these samples, we can construct the empirical CDF of the null distribution, and consequently correctly calculate the p-value.

### C.2. Number of Categories and Analysis of Type-I error and Power

The proposed method can handle any level of discretization, as long as it is greater than 1, with Type-I errors properly controlled. At the same time, more levels are always beneficial, because it leads to less information loss during the discretization process, and thus the correlation matrix can be more efficiently estimated for building the test.

Regarding Type-I errors, as we establish the exchangeability even in the discretized scenario, the asymptotic null distribution can be estimated by random permutations. Consequently, Type-I errors can be properly controlled at any significance level. At the same time, we do not have theoretical result on the analysis of the power yet. To be specific, even without

considering discretization, the analysis of power involves tools from advanced random matrix theories and is highly nontrivial. Furthermore, in our setting with discretized variables, the involved maximum likelihood step makes such an analysis even more challenging. To our best knowledge, there is not any existing result available for the analytic form of the power in our setting, and we plan to leave it for future exploration.

## D. Related Work

**Conditional independence and rank test.** A line of conditional independence tests imposes simplifying assumptions on the distributions. For instance, when the variables have linear relations with additive Gaussian noise, the Fisher's classical z-test based on partial correlations can be used (Fisher, 1924; Baba et al., 2004). Ramsey (2014) developed an approach that separately regresses $X$ and $Y$ on $Z$, and further perform independence test on the corresponding residuals. Fukumizu et al. (2007) proposed a conditional independence test method based on Hilbert-Schmidt independence criterion (HSIC) (Gretton et al., 2007). Zhang et al. (2012) further provided a kernel-based conditional test that yields pointwise asymptotic level control. Shah & Peters (2018) investigated the hardness of conditional independence test, and developed a method based on kernel-ridge regression and generalised covariance measure. On the other hand, existing statistical tests for rank of a cross-covariance matrix (Anderson, 1984) often rely on CCA (Jordan, 1875; Hotelling, 1992), with a likelihood ratio based test statistics. Recently, Sun et al. (2025b) also establishes a valid partial correlation test in the presence of discretization, with a focus on the binary discretization scenario, and later Sun et al. (2025a) better solves this problem with general method of moment.

**Permutation test.** Research and applications related to permutation tests have addressed increased attention in recent years (David, 2008; Pesarin & Salmaso, 2010; Welch, 1990). These tests lead to valid inferences while requiring weak assumptions that are commonly satisfied, base on the exchangeability of observations under the null hypothesis. Recently, a permutation-based CI test was proposed (Doran et al., 2014) and more recently a permutation-based rank test (Winkler et al., 2020). However, they cannot deal with the discretization problem. In contrast, our MPRT can take all continuous, partially discretized, or all discretized data as input, and our Type I errors can be properly controlled.

**Constraint-based causal discovery.** Constraint-based methods leverage statistical tests, such as conditional independence tests, to estimate the causal structure. Spirtes & Glymour (1991) proposed the PC algorithm that estimates the skeleton and orient certain edges to identify the Markov equivalence class. FCI (Spirtes et al., 1995; Colombo et al., 2012) was developed to allow for latent and selection variables, while the CCD algorithm (Richardson, 1996) can accommodate cycles. Furthermore, Huang et al. (2020) developed a constraint-based method that allows for heterogeneity or non-stationarity in the data distribution, while Silva et al. (2006); Huang et al. (2022); Dong et al. (2024a) proposed algorithms based on rank test that recover the causal structure involving latent confounders.

