# OpenReview forum: "Permutation-based Rank Test in the Presence of Discretization and Application in Causal Discovery with Mixed Data"
_ICML.cc/2025/Conference — ICML 2025 poster_

### Official Review · Reviewer_QSUp · 2025-03-08

**Overall Recommendation:** 4

**Summary:**

This paper introduces the Mixed data Permutation-based Rank Test (MPRT) for testing the rank of cross-covariance matrices in the presence of discretized variables. The authors address a critical gap in existing rank tests, which assume continuous measurements and fail when variables are discretized. MPRT leverages permutation-based resampling and maximum likelihood correlation estimation to handle mixed data. Theoretical guarantees on Type I error control and empirical validation on synthetic and real-world datasets demonstrate its effectiveness. The method is also applied to causal discovery, showing improved performance over traditional conditional independence tests in discretized settings.

**Claims And Evidence:**

Yes.

**Essential References Not Discussed:**

No.

**Experimental Designs Or Analyses:**

Yes, I checked the experiments in Section 4.

**Methods And Evaluation Criteria:**

Yes.

**Other Comments Or Suggestions:**

None.

**Other Strengths And Weaknesses:**

Strength: This paper proposes a valid rank test for discretized data, addressing limitations of classical CI tests that fail under discretization.

Weakness:  The theoretical guarantees rely on joint Gaussianity. While Appendix C.2 briefly discusses non-Gaussian cases, robustness to severe non-Gaussianity or nonlinear relationships is unclear.

**Questions For Authors:**

1. Can discretization create or break CI relationships, especially when the data is non-Gaussian or exhibits nonlinear dependencies? Specifically, are there cases where CI holds before discretization but not after, or vice versa?

2. It seems that the threshold estimation in Equation (14) assumes the underlying variables follow a standard Gaussian distribution. How might deviations from this assumption affect the validity of correlation estimates or rank test results?

**Relation To Broader Scientific Literature:**

This paper utilizes the permutation strategy to extend classical rank tests to discretized data.

**Theoretical Claims:**

No.

---

> ### Author Rebuttal · Authors · 2025-03-31
>
> Thank you for the time dedicated to reviewing our paper, the insightful comments, and valuable feedback. Please see our point-by-point responses below.
>
>
> **Q1:**  While Appendix C.2 briefly discusses non-Gaussian cases, robustness to severe non-Gaussianity or nonlinear relationships is unclear.
>
> **A1:**
> To the best of our knowledge, under the linear Gaussian assumption,
> none of the existing rank tests can properly handle the issue of  discretization and thus we believe our method is already a non-trivial contribution to this field. At the same time, we totally agree that non-Gaussiani or nonlinear cases can also be highly relevant in practical applications. Extending our framework to accommodate non-Gaussian cases is certainly feasible, as discussed in Appendix C.2. Addressing nonlinear cases is certainly more challenging because, even without discretization, the rank constraints (i.e., vanishing determinant constraints implied by trek separation) relies on linearity; extending this to arbitrary nonlinear relationships remains an open problem in the field of causal discovery. We thank the reviewer for the insightful comments and we plan to leave it for future exploration.
>
>
> **Q2:**  Can discretization create or break CI relationships?
>
> **A2:** Yes, discretization generally breaks CI relations regardless of the existence of non-Gaussianity or non-linearity (as long as we are not referring to the trivial case where variables are discretized into just one value, resulting in complete loss of information).
> For example, consider continuous variables $\mathsf{X},\mathsf{Y},\mathsf{Z}$ following a causal graph $\mathsf{X}\leftarrow\mathsf{Z}\rightarrow\mathsf{Y}$.
> By d-separation we know that $\mathsf{X} \perp  \mathsf{Y} | \mathsf{Z}$. However, if we can only observe a discretized version of $\mathsf{Z}$, say $\mathsf{Z}'$, the CI relation  $\mathsf{X} \perp  \mathsf{Y} | \mathsf{Z}'$ generally does not hold. The reason is that discretization from $\mathsf{Z}$ to $\mathsf{Z}'$ introduces information loss and thus $\mathsf{Z}'$ no longer retains the complete information to explain the dependence between $\mathsf{X}$ and $\mathsf{Y}$.
>
> The same reasoning also applies to the rank test. Roughly speaking, a lower rank of the cross-covariance indicates greater independence between two sets of variables.  The presence of discretization generally induces higher rank than the correct one, leading to a false indication of dependence and thus breaks the independence. Foe example, as shown in in Figure 1 of our submission,
>   panel (a) shows the population cross-covariance without discretization, where the rank is 1, while panel (b) shows the cross-covariance computed from discretized observations, resulting in a rank of 3.
>
>
> **Q3:**  It seems that Equation 14 assumes the underlying variables follow a standard Gaussian distribution. How might deviations from this assumption affect the validity of correlation estimates or rank test results?
>
> **A3:** Violation of this assumption, e.g., shifting and rescaling, does not affect the validity of the whole method, as long as all the variables are still jointly Gaussian.
> This is because we care about the rank of the cross-covariance matrix, which is equal to  the rank of the cross-correlation matrix; the latter is clearly invariant to shift or rescaling of either some or all variables (also mentioned in line 229). Thus, we can simply assume that all variables are standardized for Equation 14.
>
>
> We genuinely appreciate the reviewer's effort and hope that your concerns/questions are addressed.

---

### Official Review · Reviewer_vw91 · 2025-03-14

**Overall Recommendation:** 3

**Summary:**

The authors consider the task of finding the rank of cross covariance matrices, when some variables have been discretised. The authors consider a permutation based rank test that is able to handle both discrete and continuous variables. They show that their method is able to perform well with both continuous and mixed data.

## update after rebuttal
I thank the authors for their response. I will keep my positive score.

**Claims And Evidence:**

The claims made in the paper are supported.

**Essential References Not Discussed:**

Not to my knowledge.

**Experimental Designs Or Analyses:**

- The PC algorithm outputs the CPDAG, not a single DAG, but its Markov equivalence class. As such, the SHD and F1 scores might not make sense. The SHD between CPDAGs might be a better metric. If this is the actual metric that is calculated, it should be made clear.

**Methods And Evaluation Criteria:**

I can't find any information about the data generation for the experiments. It would be useful to see the performance under varying data generating assumptions and varying degrees of discretisation.

**Other Comments Or Suggestions:**

N/A

**Other Strengths And Weaknesses:**

Strengths:
- The paper tackles an important problem of discretisation.

Weaknesses:
- Lack of information on the experiments makes it hard to judge whether they were comprehensive.
- Although they tackle the problem of discretisation, they only do this for rank tests. Causal discovery has frameworks that can handle both mixed and continuous data. For example [1] is tested on both continuous and discrete data.

[1] Lopez-Paz, David, Philipp Hennig, and Bernhard Schölkopf. "The randomized dependence coefficient." Advances in neural information processing systems 26 (2013).

**Questions For Authors:**

- I don't understand the arguement in L241 LHS, why can Lemma 4 not be used for discrete values?
- Section 4.2: Just to be clear, CCART-C when applied on the mixed data, is not actually fed mixed data, but works on the original continuous values? If so, it might also be interesting to see what happens when mixed data is provided to CCART-C.
- What is the ground truth for section 4.4?

**Relation To Broader Scientific Literature:**

The main contribution is a permutation based rank test that can also handle discrete data.

It is not clear which how other parts are the contributions of the paper. Is the estimation of the correlation with discretisation novel (section 3.3)? It seems to follow known results.

**Theoretical Claims:**

Theoretical claims seem correct.

---

> ### Author Rebuttal · Authors · 2025-03-31
>
> Thank you for the time dedicated to reviewing our paper, the insightful comments, and valuable feedback. Please see our point-by-point responses below.
>
> **Q1:** About data generation for experiments?
>
> **A1:**
> We assume that data is generated following a linear structural causal model $\mathsf{V}\_i=\sum \nolimits\_{\mathsf{V}\_j \in \text{Pa}(\mathsf{V}\_i)} f\_{j,i} \mathsf{V}\_j + \epsilon\_{\mathsf{V}\_i}$, where each edge coefficient $f_{j,i}$ is uniformly sampled from $[-2, 2]$ and the noise terms are Gaussian with variance uniformly from $[1,5]$.
> Within this model, the population covariance can be directly specified once $f\_{j,i}$ and the variance of $\epsilon\_{\mathsf{V}\_i}$ are specified.
> %
> Information about the process of discretization can be found in line 369-373. We thank the reviewer and have revised our paper to include the above description.
>
>
> **Q2:** The metric used in Table 1?
>
> **A2:** As mentioned in line 425, the metrics used in Table 1 are SHD and F1 scores for the skeleton, which is also commonly used in the literature. In light of your suggestion, we conducted additional experiments to compare the CPDAGs in terms of SHD and F1, and our proposed MPRT still consistently outperforms other methods across different sample sizes. We thank the reviewer for the valuable comment and have revised our paper to include this additional result.
>
> **Q3:** Although they tackle the problem of discretisation, they only do this for rank tests. Causal discovery has frameworks that can handle both mixed and continuous data. For example [1] is tested on both continuous and discrete data?
>
> **A3:**
> Thank you for mentioning this interesting work and we have revised our manuscript to add a related discussion.
> From one perspective, since conditional independence constraint (in the linear case) is a special case of rank constraint of covariance matrices [3], the proposed method can also be used to test CI in the presence of discretization, as empirically validated by the results in Table 1. From another perspective,
> as you mentioned, there are some existing works that can handle mixed data for CI, e.g., RDC [1] and KCI [2]. In light of your suggestion, we conducted additional experiments to compare with KCI in Table 1 (it is hard to compare to RDC as it can only measure independence rather than conditional independence). We observe that the proposed MPRT outperforms KCI by a clear margin, e.g., when $N=2000$, MPRT achieves a F1 score of 0.96 while KCI achieves only 0.86. This outcome is expected, as KCI (like other CI tests designed for mixed data) directly test the relations between observed variables, while the real objective is to identify the relations between the underlying continuous variables. Please kindly refer to our response to Q2 for reviewer QSUp for a related discussion.
>
>
> **Q4:** Regarding the arguement in L241 LHS, why can Theorem 4 not be used for discrete values?
>
> **A4:** There are two reasons. (i) Directly using the ordinal values cannot produce the correct linear transformation $A$ (does not converge to the true one). (ii) Even if we can produce the correct $A$, when some columns of data are just ordinal values, after the linear transformation the entries of $\mathbf{C_X}_{k:}$ would be linear combination of some continuous and some ordinal values, which can be arbitrary (as the choice of ordinal values can take either $\{1,2,3\}$ or $\{1,2,10^{10}\}$) and does not have psychical meaning.
>
> **Q5:** Regarding Section 4.2, just to be clear, CCART-C when applied on the mixed data, is not actually fed mixed data, but works on the original continuous values? If so, it might also be interesting to see what happens when mixed data is provided to CCART-C.
>
> **A5:** Yes, CCART-C takes the original continuous values as input and thus in our experiments its performance serves as the upper-bound. When CCA-based rank test takes the mixed data as input, it is named as CCART-D in our experiments (as mentioned in section 4.1). As expected, CCART-D
> cannot properly control the type-I error and it does not benefit from the increase of the sample size, as shown in Figure 2.
>
> **Q6:** The ground truth for section 4.4?
>
> **A6:** The structure that underlies human personality remains an open research problem and thus there is no established ground truth for it yet.
>
> We genuinely appreciate the reviewer's effort and hope that your concerns/questions are addressed.
>
> [1] Lopez-Paz, The randomized dependence coefficient. 2013.
>
> [2] Zhang, Kernel-based conditional independence test and application in causal discovery. 2012.
>
> [3] Sullivant, Trek separation for gaussian graphical models. 2010.

---

### Official Review · Reviewer_uits · 2025-03-15

**Overall Recommendation:** 3

**Summary:**

This paper introduces the Mixed Data Permutation-Based Rank Test (MPRT), an approach designed to address the challenge of discretization in rank tests. The proposed MPRT estimates the asymptotic null distribution by leveraging data permutation, which is grounded in the exchangeability condition of the linear projection of the relevant variables. The authors demonstrate the efficacy of the MPRT through comprehensive synthetic and real-world experiments.

**Claims And Evidence:**

No. The correctness of Thm. 4, which is the key result that the proposed test relies on, is problematic.

First, there is no clear definition of "asymptotic independence". There are various definitions in the literature [1,2], but it is unclear which definition the authors adopt. From the proof, I presume that the authors define two sequence of random vectors $X_n$ and $Y_n$ are asymptotically independent if there exists $X, Y$ such that $X \perp Y$ and $X_n \to X, Y_n \to Y$ in distribution [1]. But this needs clarification.

Second, the asymptotic independence between $C_{X k:}$ and $C_{Y k:}$ is not discussed in a rigorous way. The authors should follow the definition and show the convergence of  $C_{X k:}, C_{Y k:}$ to some $C^\prime_{X}, C^\prime_{Y}$ such that $C^\prime_{X} \perp C^\prime_{Y}$.

From the proof, I conjecture that the authors may want to show this by first showing the convergence of $\hat{\Sigma}_X, \hat{\Sigma}_Y$ to $\Sigma_X, \Sigma_Y$, and then using the continous mapping theorem to show the convergence of C{X k:}, C{Y k:}, which are computed from $\hat{\Sigma}_X, \hat{\Sigma}_Y$ with SVD decomposition.

But this can be highly problematic:

1. The convergence of $\hat{\Sigma}_X, \hat{\Sigma}_Y$ obtained from pseudo-maximum likelihood is unkown. I do not find results in the paper cited by the authors (Besag, 1974).

2. The SVD is not unique and thus cannot be considered as a continuous function (see [3]). So the continous mapping theorem may not apply here.

Can the authors provide further clarifications to my questions? I ask for particular rigor because the proposed test highly rely on this independence result. I am happy to increase my score if the authors can address these questions properly.

[1] https://math.stackexchange.com/questions/1272661/is-there-a-concept-of-asymptotically-independent-random-variables

[2] https://arxiv.org/pdf/1910.04243

[3] https://math.stackexchange.com/questions/3389899/continuity-of-singular-value-decomposition

**Essential References Not Discussed:**

Yes.

**Experimental Designs Or Analyses:**

Yes. The experiment is comprehensive.

**Methods And Evaluation Criteria:**

The permutation-based test relies on the exchangeability result (Thm. 4), which can be problematic (see Claims and Evidence). Therefore, the proposed test may not be valid.

**Other Comments Or Suggestions:**

See Claim and Evidence.

**Other Strengths And Weaknesses:**

1. The problem of discretization in rank-based test is important.

2. The writting is clear.

**Questions For Authors:**

See Claim and Evidence.

**Relation To Broader Scientific Literature:**

This paper extends the classifical rank test (Jordan, 1875; Hotelling, 1992) to cases with discretization.

**Theoretical Claims:**

Yes. The proof of Thm. 4 is problematic. See Claim and Evidence.

---

> ### Author Rebuttal · Authors · 2025-03-31
>
> Thank you for your insightful comments, which have greatly helped us refine the quality of our paper. We first provide responses to your concerns and then give an updated sketch of proof for Thm 4.
>
> **Q1:** The convergence of $\hat{\Sigma}\_\mathbf{X}$ by pseudo-maximum likelihood.
>
> **A1:** Yes, $\hat{\Sigma}\_\mathbf{X}$ in our method converges in probability to the population one. [1] is one of the earliest on pseudo-likelihood, and its consistency was later discussed in [2]. As a member of M-estimators, the  consistency can also be derived from the general theory for M-estimators [3]. Specifically in the presence of discretization, [6] derived the consistence of estimating precision matrix under mixed non-paranormal model; similar arguments can be applied for the consistency of $\hat{\Sigma}\_\mathbf{X}$ by pseudo-maximum likelihood in our paper.
>
> **Q2:** Continuity and uniqueness of SVD.
>
> **A2:** SVD is  not continuous only when the input matrix has repeated singular values. Specifically, if a matrix $A$ has distinct singular values, then SVD is continuous in the neighborhood of $A$, and unique only up to sign flip (chapter 2 section 5.3 of [4]).
> Thus, to make use of the continuous mapping theorem, we  assume that $\Sigma\_\mathbf{X}^{-\frac{1}{2}}\Sigma\_{\mathbf{X},\mathbf{Y}}\Sigma\_\mathbf{Y}^{-\frac{1}{2}}$ does not have repeated singular values (the justification of which is discussed in Q3). To further eliminate the sign indeterminacy, we can follow scikit-learn to ensure the largest coefficient in each column, in terms of absolute value, is positive (svd flip in scikit-learn).
>
> **Q3:** The justification of assuming non-repeated singular values.
>
> **A3:** Faithfulness is one of the most important assumptions in causal discovery and is typically justified by that the set of parameters violating it is of Lebesgue measure zero [7].
> Similarly, it has been shown that the set of matrices with repeated singular values has Lebesgue measure zero (Lemma 1.4.2 in [8], also in [5]).
>
> An updated sketch of proof for Thm 4.
>
> > Sketch of proof
> \
> Given $(\hat{\Sigma}\_\mathbf{X},\hat{\Sigma}\_\mathbf{Y},\hat{\Sigma}\_{\mathbf{X},\mathbf{Y}})\overset{p}{\to}(\Sigma\_\mathbf{X},\Sigma\_\mathbf{Y},\Sigma\_{\mathbf{X},\mathbf{Y}})$,
> we aim to show the desired asymptotic independence. Specifically we want to show (i) $\mathbf{C_X}\_{k:} \overset{p}{\to} \mathbf{C_X}\_{k:}^\*$  and $\mathbf{C_Y}\_{k:} \overset{p}{\to} \mathbf{C_Y}\_{k:}^\*$, and  (ii) $\mathbf{C_X}\_{k:}^\*,\mathbf{C_Y}\_{k:}^\*$ are independent under the null hypo.
> \
> Here $\mathbf{C_X}=A^T\mathbf{X},\mathbf{C_Y}=B^T\mathbf{Y}$,
> $\mathbf{C_X}^\*={A^\*}^T\mathbf{X}$, and $\mathbf{C_Y}^\*={{B^\*}^T}\mathbf{Y}$, where $(A,B)$ and $(A^\*,B^\*)$  are produced by SVD using  estimated covariance and population one respectively as follows.
> $$USV=\hat{\Sigma}\_\mathbf{X}^{-\frac{1}{2}}\hat{\Sigma}\_{\mathbf{X},\mathbf{Y}}\hat{\Sigma}\_\mathbf{Y}^{-\frac{1}{2}},A=\hat{\Sigma}\_\mathbf{X}^{-\frac{1}{2}T}U,B=\hat{\Sigma}\_\mathbf{Y}^{-\frac{1}{2}T}V^T,~~~U^\*S^\*V^\*  =\Sigma\_\mathbf{X}^{-\frac{1}{2}}\Sigma\_{\mathbf{X},\mathbf{Y}} \Sigma\_\mathbf{Y}^{-\frac{1}{2}},A^\*=\Sigma\_\mathbf{X}^{-\frac{1}{2}T}U^\*,B^\*=\Sigma\_\mathbf{Y}^{-\frac{1}{2}T}{V^\*}^{T}.$$
> For (i): By continuous mapping theorem, under the assumption of no repeated singular values, we have $U \overset{p}{\to} U^\*$. As $\Sigma\_\mathbf{X}$ is positive definite, the matrix inverse square root is continuous and thus
> $\hat{\Sigma}\_\mathbf{X}^{-\frac{1}{2}T}\overset{p}{\to}\Sigma\_\mathbf{X}^{-\frac{1}{2}T}$. Given $(U,\hat{\Sigma}\_\mathbf{X}^{-\frac{1}{2}T})\overset{p}{\to}(U^\*,\Sigma\_\mathbf{X}^{-\frac{1}{2}T})$, we have $\hat{\Sigma}\_\mathbf{X}^{-\frac{1}{2}T}U=A\overset{p}{\to}A^\*=\Sigma\_\mathbf{X}^{-\frac{1}{2}T}U^\*$. Similarly, we have $B\overset{p}{\to}B^\*$.
> \
> Thus $$((A^T-{A^\*}^T)\mathbf{X},(B^T-{B^\*}^T)\mathbf{Y})\overset{p}{\to}0\Rightarrow (((A^T-{A^\*}^T)\mathbf{X})\_{k:},((B^T-{B^\*}^T)\mathbf{Y})\_{k:})\overset{p}{\to}0\Rightarrow(\mathbf{C_X}\_{k:},\mathbf{C_Y}\_{k:})\overset{p}{\to}(\mathbf{C_X}\_{k:}^\*,\mathbf{C_Y}\_{k:}^\*).$$
> For (ii): Under the null hypo, the cross-covariance between $\mathbf{C_X}\_{k:}^\*$ and $\mathbf{C_Y}\_{k:}^\*$ are all zeros.
> As $\mathbf{C_X}\_{k:}^\*,\mathbf{C_Y}\_{k:}^\*$ are jointly gaussian (linear mixing of $\mathbf{X,Y}$),
> zero cross-covariance implies independence.
>
> Please feel free to let us know if any part remains unclear or you have further questions. Thank you again for your valuable feedback!
>
> [1] Besag, Spatial interaction ... system. 1974.
>
> [2] Gourieroux, Pseudo maximum likelihood methods: Theory. 1984.
>
> [3]  Gourieroux, Consistent pseudo ... estimators. 2017.
>
> [4] Kato, Perturbation theory for linear operators. 2013.
>
> [5] Bochnak, Real algebraic geometry.2013.
>
> [6] Fan, High dimensional ... mixed data. 2017.
>
> [7] Spirtes, Causation, prediction, and search. 2000.
>
> [8] Kunisky, Lecture Notes on Random Matrix Theory. 2024.

---

### Decision · Program_Chairs · 2025-05-01

**Decision:**

Accept (poster)

**Comment:**

This paper proposes the Mixed Data Permutation-Based Rank Test (MPRT) to assess the rank of cross-covariance matrices when data includes both discrete and continuous variables.


Pros:

+ Addresses a critical limitation in classical rank tests by effectively handling discretized and mixed data.

+ Clear writing and solid motivation; the application to causal discovery adds practical relevance.

Cons:

+ The key theoretical result (Theorem 4) needs clarification and refinement, especially regarding asymptotic independence and SVD continuity.

+ Experimental details (e.g., data generation, metrics, and assumptions) are insufficient in parts, making reproducibility and evaluation difficult.

+ Robustness to non-Gaussianity and nonlinear dependencies is underexplored, which may limit the method's applicability in real-world scenarios.